# A Comprehensive Taxonomy for Prediction Models in Software Engineering

**Xinli Yang, Jingjing Liu**  **and Denghui Zhang** *

College of Information Science and Technology, Zhejiang Shuren University, Hangzhou 310015, China
* Correspondence: dhzhang@zjsru.edu.cn

**Abstract:** Applying prediction models to software engineering is an interesting research area. There have been many related studies which leverage prediction models to achieve good performance in various software engineering tasks. With more and more researches in software engineering leverage prediction models, there is a need to sort out related studies, aiming to summarize which software engineering tasks prediction models can apply to and how to better leverage prediction models in these tasks. This article conducts a comprehensive taxonomy on prediction models applied to software engineering. We review 136 papers from top conference proceedings and journals in the last decade and summarize 11 research topics prediction models can apply to. Based on the papers, we conclude several big challenges and directions. We believe that the comprehensive taxonomy will help us understand the research area deeper and infer several useful and practical implications.

**Keywords:** artificial intelligence; prediction model; software engineering; comprehensive taxonomy

## 1. Introduction

Software engineering is an important research area. It aims to help software developers in terms of time and effort, and improve software quality in terms of stability, reliability and security. There have been a large number of software engineering studies, which cover various research topics such as defect prediction, bug report management and etc.

Prediction model is an important technique, which can be applied to many different research areas. Specifically, a prediction model can be built based on training data for prediction, classification, identification and detection (such as text classification, image identification and malware detection). There are various prediction models which leverage many different machine learning algorithms, such as naive Bayes, decision tree, support vector machine and random forest. Different prediction models have their respective pros and cons, and preform well in different research problems.

Applying prediction models to software engineering is an interesting area and can help solve many difficult problems. For example, in defect prediction researchers aim to recommend software components that are likely to be defective to developers, which can reduce time and effort cost of developers in software debugging, improve the efficiency of development process and guarantee software quality. In performance prediction, researchers aim to predict the performance of a system, which is crucial in practice. There have been a large number of related studies in the last decade [1]. These studies have achieved great improvement in various software engineering tasks. Although prediction models contribute much to software engineering, there is a lack of a systematic taxonomy about to which software engineering tasks prediction models can apply and how to well leverage prediction models in these tasks.

To bridge the gap, in the paper we conducted a comprehensive taxonomy on prediction models applied in software engineering. We covered 136 papers from three top conference proceedings (i.e., ICSE, FSE and ASE) and two top journals (i.e., TSE and TOSEM) in the recent 10 years in total. We choose them because they are five most top and popular

conference proceedings and journals in the software engineering area and the papers in them are representative for the state-of-the-art software engineering researches. We found that prediction models have been applied to various software engineering tasks. Based on the software development process, we grouped all tasks into 11 main research topics, i.e., software coding aid, software defect prediction, software management, software quality, software performance prediction, software effort estimation, software testing, software program analysis, software traceability, software bug report management, software users and developers. All the topics play key roles in the software development process. By leveraging prediction models, researchers have achieved good performance in the tasks in these 11 research topics.

Based on the papers, we concluded several big challenges when applying prediction models to software engineering tasks, to which researchers shall pay much attentions in their later work. In addition, we also showed several promising research directions, through which researchers may achieve great improvement.

The main contributions of the paper are:

1.  We conduct a comprehensive taxonomy on prediction models applied to software engineering. The taxonomy contains 136 primary papers from top conference proceedings and journals in the last decade.
2.  We summarize the 136 papers into 11 main research topics. Based on them, we conclude several big challenges and promising directions when applying prediction models to software engineering tasks.

The rest of our paper is organized as follows. Section 2 introduces the basis of prediction models. Section 3 presents our research method. Sections 4–14 summarize primary studies related to prediction models applied to software engineering tasks for each of research topics. Section 15 describes the challenges and directions of prediction models applied to software engineering tasks. Conclusions are presented in the last section.

## 2. Basis of Prediction Models

In this section, we first introduce the basic concepts of prediction models in Section 2.1. Next, we briefly introduce five most common algorithms used in prediction models in Section 2.2. Finally, we present several common evaluation metrics for prediction models in Section 2.3.

### 2.1. Overview

Prediction models have been applied to various tasks in software engineering [2]. To build prediction models, there are several key elements, which are listed as follows:

1.  **Datasets.** Datasets are the input of prediction models. There are various datasets (such as code, bug reports). Different software engineering tasks usually have different datasets which have different properties (such as scale, distribution, bias and etc). Due to these reasons, different prediction models are needed to fit well for different datasets.
2.  **Features.** A dataset contains more or fewer features. Features play a crucial role in building prediction models. A good feature set can generate a prediction model with very good performance, while a weak feature set may lead to a useless prediction model.
3.  **Algorithms.** There are various prediction models and their key difference lies in the algorithms. There are many algorithms in prediction models and different algorithms may fit different software engineering tasks. Section 2.2 introduces several common algorithms used in prediction models in detail.
4.  **Evaluation Metrics.** When prediction models output their prediction results, we use metrics evaluating their effectiveness so that we can pick up the best prediction model for a specific software engineering task. Similarly, there are many evaluation metrics for prediction models and different metrics may fit different software engineering

tasks. Section 2.3 introduces several widely-used evaluation metrics for prediction models in detail.

### 2.2. Common Algorithms

There are many algorithms for prediction models. Different algorithms fit to different problems and no one algorithm can always perform the best. In the section we introduce five most common algorithms, i.e., naive Bayes (NB), random forest (RF), logistic regression (LR), support vector machine (SVM) and k-nearest neighbors (KNN). We introduce these algorithms since they are widely used in software engineering tasks and they often appear as baselines in software engineering papers [3]. Moreover, they are very classic among all the prediction models [3].

#### 2.2.1. Naive Bayes

Naive Bayes (NB) is a probabilistic model based on Bayes theorem for conditional probabilities [2,4]. Naive Bayes assumes that features are independent from one another. Moreover, all the features are binominal. That is, each feature only has two values of 0 and 1.

Based on the above assumptions, given a feature vector $V = (f_1, f_2, \ldots, f_n)$ ($f_i$ represents a specific feature) and a label $l_j$, the probability of the $V$ given the label $l_j$ is:

$$p(V|L = l_j) = \prod_{i=1}^{n} p(f_i|L = l_j).$$

With Bayes' theorem, we can compute the probability of a label $l_j$ given the $V$. Since Naive Bayes assumes that features are independent from one another, the equation can be written as:

$$p(L = l_j|V) = \prod_{i=1}^{n} p(f_i|L = l_j).$$

The probability of the feature $f_i$ given class $l_j$ (i.e., $p(f_i|L = l_j)$) in the above equation can be estimated based on the training data. Next, based on the above equation, we can compute the probability for every label given an unlabeled feature vector $V$, and assign the label with the highest probability to it.

There are several variants of NB, one of which is called Naive Bayes Multinominal (NBM) [2,4]. NBM is very similar to NB. However, in NBM the value of each feature is not restricted to 0 or 1, rather it can be any non-negative number. NBM and NB have different advantages. For the problems that have a large number of features, NB has much smaller feature space than NBM. Therefore, in the case NB is often better than NBM. On the contrary, for the problems that do not have many features, NBM often performs better since it express features in a finer granularity.

#### 2.2.2. Random Forest

Random Forest is an advanced bagging technique based on decision tree [2,4].

Decision tree is modeled with the use of a set of hierarchical decisions on the feature variables, arranged in a tree-like structure [2,4]. In the tree-constructing process, Decision tree can rapidly find the feature variables that differentiate different classes the most. In addition, it can generate explicit rules for different classes, while many other classifiers cannot.

Bagging works best when the base learners are independent and identically distributed. However, traditional decision trees constructed using bagging cannot meet this condition. Random Forest solve the problem by introducing randomness into the model building process of each decision tree. In the construction of traditional decision trees, the split of each node are performed by considering the whole set of features, while in random forest,

the splits in each tree are performed by considering only a random subset of all features. The randomized decision trees have less correlation so that bagging them performs better.

### 2.2.3. Logistic Regression

Logistic regression (LR) is a kind of generalized linear model [2,4]. There are two key differences between Logistic regression and linear regression. First, Logistic regression assumes data are in the Bernoulli distribution. Second, Logistic regression predicts the probability of particular outcomes through the logistic distribution function. A logistic function $L$ is a sigmoid curve with the following equation:

$$L = \frac{1}{1 + e^{-WV}}.$$

In the equation, $W$ is the parameter vector used to combine different features in $V$.

### 2.2.4. Support Vector Machine

Support Vector Machine (SVM) is developed from traditional linear models [2,4]. As with all traditional linear models, it uses a separating hyperplane as the decision boundary to differentiate two classes. Given training data (i.e., feature vectors), SVM first maps each feature vector to a point in a high-dimensional space, in which each feature represents a dimension. Then, SVM selects the points which have big impact for classification as support vectors. Next, it creates a separating hyperplane as a decision boundary to classify two classes. When an unlabeled data instance needs to be classified, SVM can assign it a label according to the decision boundary. Compared with traditional linear models, SVM considers structural error which includes both empirical error and confidence error. Therefore, the separating hyperplane created by SVM has a maximum margin (i.e., it separates the support vectors belonging to the two classes as far as possible), which makes SVM one of the best classifiers.

### 2.2.5. K-Nearest Neighbors

K-nearest neighbors (KNN) is an instance-based classifier [2,4]. Its principle is intuitive: similar instances have similar class labels. In our setting, KNN mainly contains three steps. First, similar to SVM, KNN maps all the training data to points in a high-dimensional space. Then, for an unlabeled feature vector $V$, we find $k$ most nearest points to it based on a specific distance metric. There are various distance metrics, such as Euclidean distance and Manhattan distance. Finally, we determine the label of $V$ by the labels of the majority of its $k$ nearest neighbors.

### 2.3. Evaluation Metrics

To evaluate the effectiveness of a prediction model, there are several widely-used evaluation metrics, i.e., precision, recall, F1-score, AUC (Area Under the Receiver Operating Characteristic Curve). Many software engineering studies use them as evaluation metrics [1,5,6].

The evaluation metrics can be derived from a confusion matrix, as shown in Table 1. The confusion matrix lists all four possible prediction results. If a data instance is correctly predicted as positive, it is a true positive (TP); if a data instance is wrongly predicted as positive, it is a false positive (FP). Similarly, there are false negatives (FN) and true negatives (TN). Based on the four numbers, the evaluation metrics can be calculated.

**Table 1.** Confusion Matrix.

|  | **Predicted Positive** | **Predicted Negative** |
|---|---|---|
| Truly Positive | TP | FN |
| Truly Negative | FP | TN |

### 2.3.1. F1-Score

Precision is the ratio of the number of correctly predicted as positives to the total number of predicted positives ($P = \frac{TP}{TP+FP}$). Recall is the ratio of the number of correctly predicted as positives to the actual number of positives ($R = \frac{TP}{TP+FN}$). Finally, F1-score is a summary measure that combines both precision and recall ($F = (2 \times P \times R)/(P + R)$). F1-score evaluates if an increase in precision (recall) outweighs a reduction in recall (precision). The larger F1-score is, the better is the performance of a classification algorithm. F1-score ranges from 0 to 1, with 1 representing perfect prediction performance [1,6,7].

### 2.3.2. AUC

To compute AUC, we first plot the Receiver Operating Characteristic Curve (ROC). ROC is a plot of the true positive rate (TPR) versus false positive rate (FPR). TPR is the ratio of the number of correctly predicted as positives to the actual number of positives ($TPR = \frac{TP}{TP+FN}$). FPR is the ratio of wrongly predicted as positives to the actual number of negatives ($FPR = \frac{FP}{FP+TN}$). With the ROC, AUC can be calculated by measuring the area under the curve. AUC measures the ability of a classification algorithm to correctly rank positives and negatives. The larger the AUC is, the better is the performance of a classification algorithm. The AUC score ranges from 0 to 1, with 1 representing perfect prediction performance. Generally, an AUC score above 0.7 is considered reasonable [1,6,7].

## 3. Research Methods

To conduct a comprehensive taxonomy, we follow a systematic and structured method inspired by other reviews and taxonomies [8–12].

### 3.1. Paper Sources and Search Strategy

To conduct a comprehensive taxonomy, we investigate the papers in the last decade from five datasets, which contain three top conference proceedings (i.e., ICSE (International Conference on Software Engineering), FSE (International Symposium on the Foundations of Software Engineering) and ASE (International Conference on Automated Software Engineering)) and two top journals (i.e., TSE (Transactions on Software Engineering) and TOSEM (Transactions on Software Engineering and Methodology)).

Note that the papers in the five datasets can be related to any topics in software engineering, and in this taxonomy, we focus on the papers about prediction models. To do so, we adopt a search strategy which mainly contains three steps, i.e., paper collection, automated filtering, manual filtering. We will elaborate each step in the following text.

**Paper Collection.** First, we download all the papers of the five datasets based on dblp computer science bibliography (https://dblp.org/, accessed on 6 Feburary 2023), which is an on-line reference for bibliographic information on major computer science publications and provides open bibliographic information on major computer science journals (about 32,000 journal volumes) and proceedings (more than 31,000 conference or workshop proceedings). DBLP indexes all the papers of the five datasets, and for each paper, it has links from either IEEE *Xplore* (The IEEE (Institute of Electrical and Electronics Engineers) *Xplore* Digital Library is a powerful resource for discovery of and access to scientific and technical content published by the IEEE and its publishing partners. It provides web access to more than three million full-text documents from some of the world's most highly cited publications in electrical engineering, computer science and electronics. The website is: http://ieeexplore.ieee.org/Xplore/home.jsp, accessed on 6 Feburary 2023) or ACM (The ACM (Association for Computing Machinery) Digital Library is a research, discovery and networking platform containing the full-text collection of all ACM publications, including journals, conference proceedings, technical magazines, newsletters and books. The website is: http://dl.acm.org/, accessed on 6 Feburary 2023) Digital Library, or both. Therefore, we can download all the papers from IEEE *Xplore* or ACM Digital Library through the links.

**Automated Filtering.** After we download all the papers of the five datasets from IEEE *Xplore* and ACM, we filter them to select the papers leveraging prediction models. The initial number of total papers is over 2000. If we look over all the papers in person, it will cost too much time and effort due to the large amount. Therefore, we first use several keywords to automatically filter all the papers. Specifically, we search whether one of the keywords appear in the title and abstract of a paper. If so, the paper is preserved, and otherwise it is filtered. The keywords we use are shown in Table 2. They are mainly grouped into two categories, i.e., target-based keywords and technique-based keywords. Note that for the search strategy the keywords are not case-sensitive and not "whole words only", which means that the stemmed keywords can represent all forms related to them (e.g., "classif" can search for "classify", "classifying", "classified" as well as "classification").

**Table 2.** The Keywords We Use In Automated Filtering.

| Target-Based | Technique-Based |
| --- | --- |
| predict | support vector machine |
| classif | decision tree |
| identif | Bayes |
| detect | machine learning |
| | regression |
| | random forest |

**Manual Filtering.** After automated filtering, we filter most papers and preserve 319 papers. For these papers, we browse them one by one manually to ensure that they are truly related to prediction models. Finally, we select 136 papers which are definitely related to prediction models.

*3.2. Statistics of Selected Papers*

Before we review the selected paper in detail, we first summarise them in terms of publication distribution, authors and research topics.

Table 3 shows the journal/proceeding distribution of the selected papers we investigate. From the table, we can see that ICSE occupies the biggest proportion, while TOSEM has the least papers (i.e., only five in the last 10 years) related to prediction models. This suggests that if researchers are studying prediction models applied to software engineering tasks, they can look for papers in ICSE in priority.

**Table 3.** The Journal/Proceeding Distribution Of The Selected Papers.

| Dataset | Number of Papers |
| --- | --- |
| ICSE | 49 |
| FSE | 29 |
| ASE | 24 |
| TSE | 29 |
| TOSEM | 5 |

Table 4 shows the top-10 co-authors on prediction models applied to software engineering tasks. From the table, we can see that among the selected papers, one author does not contribute to many papers. All the authors contribute to less than 10 papers, except for the first one, Sunghun Kim, who has 12 papers. It suggests that the author distribution is quite scattered, which is reasonable since prediction models can be applied to various software engineering tasks.

**Table 4.** Top-10 Co-authors.

| Author | Number of Papers |
|---|---|
| Sunghun Kim | 12 |
| Ahmed E Hassan | 8 |
| Thomas Zimmermann | 6 |
| Premkumar Devanbu | 6 |
| Hongyu Zhang | 5 |
| Tim Menzies | 5 |
| David Lo | 4 |
| Foyzur Rahman | 4 |
| Jaechang Nam | 4 |
| Norbert Siegmund | 4 |

We also classify the selected papers based on the main research topics. The result is shown in Figure 1. From the figure, we can see there are many different research topics that leverage prediction models and we group them to 11 main topics in total. The top-three research topics that often leverage prediction models are "Defect Prediction", "Bug Report Management" and "Coding Aid", all of which occupy over 10%, and "Defect Prediction" even occupies near one quarter of all. The other eight research topics do not have many papers (6–12 papers each). It suggests that these research topics may have large improvement space by leveraging more prediction models.

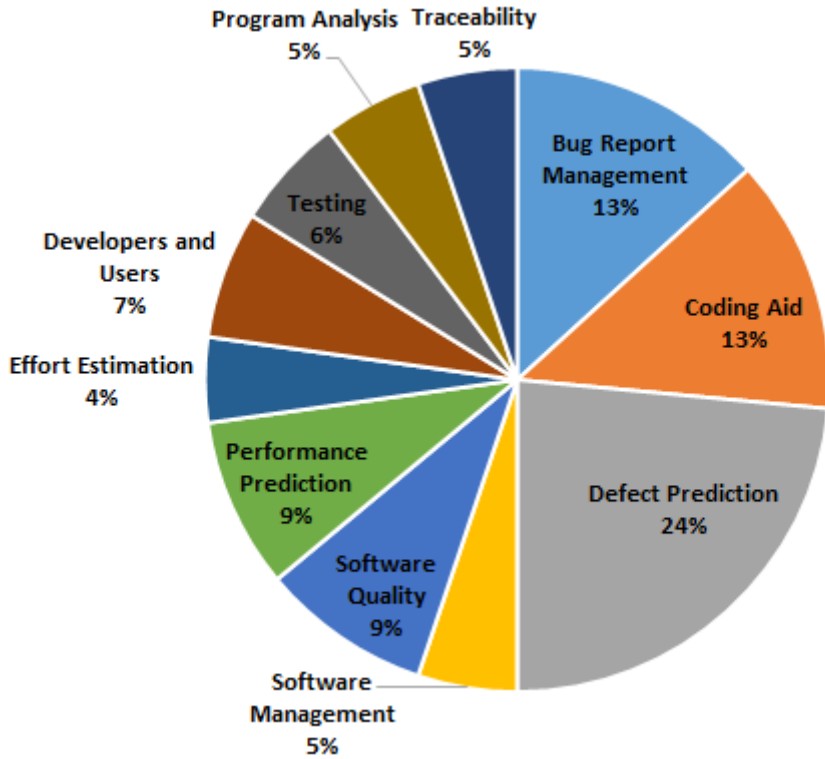

**Figure 1.** Research Topics.

In the following eleven sections, we review the selected papers in detail based on the above 11 research topics.

## 4. Coding Aid

In this section, we introduce four sub tasks in coding aid tasks, which are code development, code review, code evaluation and code API.

### 4.1. Code Development

There are many studies that aim to aid development process and improve code quality [13–19].

Reiss presented a system that uses machine learning to deduce the coding style from a corpus of code and then applies this knowledge to convert arbitrary code to the learned style [13]. They included formatting, naming, ordering, and equivalent programming constructs. They compared the Weka system (a C4.5 decision tree learner, a support vector machine, the K* instance-based learner, and a version of the RIPPER rule learner) and the Mallet system (a naive Bayes learner, a Maximum Entropy learner, and a C4.5 tree learner). Overall, the results showed that the algorithms can effectively learn ordering styles and the Weka J48 tree learner tended to do quite well. Wang et al. proposed a novel approach that automatically predicts the harmfulness of a code cloning operation at the point of performing copy-and-paste [14]. They used Bayesian Networks based on history, code and destination features. In a later work, they also used Bayesian Networks to predict whether an intended code cloning operation requires consistency maintenance [15]. Their insight is that whether a code cloning operation requires consistency maintenance may relate to the characteristics of the code to be cloned and the characteristics of its context. Therefore, they used a number of attributes extracted from the cloned code and the context of the code cloning operation. Bruch et al. proposed an advanced code completion system that recommends method calls as a synthesis of the method calls of the closest source snippet found [16]. Their algorithm is called best matching neighbors (BMN), which is based on k-nearest neighbors algorithm. They computed the distances between the current programming context and the example codebase based on the Hamming distance on a partial feature space. Proksch et al. proposed Pattern-based Bayesian Networks (PBN) for intelligent code completion [17]. They extend BMN by adding more context information and introducing Bayesian networks as an alternative underlying model. The resulting PBN model predicts the probability of methods that may appear in a specific place of code segment. Hassan et al. created decision trees to predict the certification result of a build ahead of time [18]. They mined historical information (code changes and certification results) for a large software project which is being developed at the IBM Toronto Labs. By accurately predicting the outcome of the certification process, members of large software teams can work more effectively in parallel. Zhu et al. proposed a framework, which aims to provide informative guidance on logging during development for developers [19]. They extracted structural, textual and syntactic features, used information gain to select features, and used SMOTE and CLNI to handle noises. They compared Naive Bayes, Bayes Net, Logistic Regression, Support Vector Machine (SVM), and Decision Tree and found that Decision Tree is the best to decide where to log.

### 4.2. Code Review

There are many code segments that may have defects (i.e., bugs) so that code review is a common task. Generally, code review cost much time and effort. To make code review more effective, several studies propose different methods [20–22].

Kim et al. proposed a history-based warning prioritization algorithm that mines previous fix-and-warning removal experiences that are stored in the software change history [20]. This algorithm is inspired by the weighted majority voting and Winnow online machine learning algorithms. If a warning instance from a warning category is eliminated by a fix-change, they assumed that this warning category is important. Shihab et al. used logistic regression prediction models to identify files containing high-impact defects (i.e., breakage and surprise defects) [21]. They found that the number of pre-release defects and file size are good indicators of breakage defects, whereas the number of co-changed files

and the amount of time between the latest pre-release change and the release date are good indicators of surprise defects. Padhye et al. presented NeedFeed, a system that models code relevance by mining a project's software repository and highlights changes that a developer may need to review [22]. They used both history-based and text-based features to improve the performance. However, they found that there is no clear winner between the various classification techniques.

### 4.3. Code Evaluation

For a large amount of code, researchers measure the code quality and propose several methods to comprehend code better [23–26].

Liu et al. presented search-based software quality models with genetic programming [23]. They considered 17 different machine learners in the WEKA data mining tool. They showed that the total cost of misclassification of their genetic programming based models are consistently lower than those of the non-search-based models. They selected 13 primitive software metrics (i.e., three McCabe metrics, five metrics of Line Count, four basic Halstead metrics, and one metric for Branch Count). Goues et al. measured code quality based on seven metrics (i.e., code churn, author rank, code clones, code readability, path feasibility, path frequency and path density) using linear regression [24]. They showed that measuring code quality can improve specification mining. Femmer et al. proposed a simple, yet effective approach to detect inconsistencies in wrappers [25]. They first used lightweight static analysis for extracting source code metrics, constants, etc. to compare two implementations, and then used KNN to automatically classify the differences and predict whether or not the implementations are equivalent. Rigby et al. proposed to detect which of the code elements in a document are salient or germane to the topic of the post [26]. They use decision trees to identify salient code element based on different features (TF-IDF, Element Kind, Context and Location and text type).

### 4.4. Code APIs

In the software development process, APIs play an important role. Developers often call many API classes and methods to speed coding and improve code quality. There are many studies related to code APIs [27–30].

Thummalapenta et al. proposed a code-search-engine-based approach SpotWeb that detects hotspots in a given framework by mining code examples gathered from open source repositories available on the web [27]. The hotspots are API classes and methods that are frequently reused. SpotWeb could better help developers to reuse open source code. Wu et al. proposed an iterative mining approach, called RRFinder, to automatically mining resource-releasing specifications for API libraries in the form of (resource-acquiring, resource-releasing) API method pairs [28]. In RRFinder they used decision tree to identifies resource-releasing API methods based on natural language Information, source code information, static structural information, method behavioral information and method relationship information. Petrosyan et al. proposed a technique for discovering API tutorial sections that help explain API types [29]. They classified fragmented tutorial sections using supervised text classification based on both linguistic and structural features. The technique can help developers quickly learn a subset of an API. Treude et al. presented a machine learning approach SISE to automatically augment API documentation with "insight sentences" from Stack Overflow [30]. Insight sentences can provide insight to an API type, but not contained in the API documentation of that type. SISE used as features the sentences themselves, their formatting, their question, their answer, and their authors (meta data available on Stack Overflow) as well as part-of-speech tags and the similarity of a sentence to the corresponding API documentation. They compared k-nearest neighbour, J48 decision trees, Naive Bayes, random forest, and support vector machine and found that support vector machine has the best performance in classifying whether a sentence is an insight sentence or not.

## 5. Defect Prediction

Defect prediction techniques are proposed to help prioritize software testing and debugging; they can recommend software components that are likely to be defective to developers. Rahman et al. compared static bug finders and defect prediction models [31]. They found that, in some settings, the performance of certain static bug-finders can be enhanced using information provided by statistical defect prediction models. Therefore, applying prediction models to defect prediction is a promising direction.

### 5.1. Framework

Song et al. proposed and evaluated a general framework for software defect prediction that supports unbiased and comprehensive comparison between competing prediction systems [32]. They first evaluated and chose a good learning scheme (which consists a data preprocessor, an attribute selector and a learning algorithm), and then used the scheme to build a predictor. Their framework poses three key elements for defect prediction models, i.e., datasets, features and algorithms. There are many influential studies for each of them and we will elaborate them in the next three sections.

### 5.2. Datasets

A good prediction model relies heavily on the dataset it learns from, which is also the case for defect prediction.

There are many studies investigating the quality (such as bias and size) of datasets for defect prediction [33–35]. Bird et al. used prediction models to investigate historical data from several software projects (i.e., Eclipse and AspectJ), and found strong evidence of systematic bias [33]. They concluded that bias is a critical problem that threatens both the effectiveness of processes that rely on biased datasets to build prediction models and the generalizability of hypotheses tested on biased data. Rahman et al. investigated the effect of size and bias of datasetes in defect prediction using logistic regression model [34]. They investigated 12 open source projects and their results suggested that size always matters just as much bias direction, and in fact much more than bias direction when considering common evaluation measures such as AUC and F-score. Tantithamthavorn et al. used random forest to investigate the impact of mislabelling on the performance and interpretation of defect models [35]. They found that precision is rarely impacted by mislabelling while recall is impacted much by mislabelling. In addition, the most influential features are generally robust to mislabelling.

For the datasets which have bad quality, researchers also have proposed several approaches to address them [36–38]. Kim et al. proposed an approach named CLNI to deal with the noise in defect prediction [36]. CLNI could effectively identify and eliminate noises and the noise-eliminated training sets produced by CLNI can improve the defect prediction performance. Menzies et al. applied automated clustering tools and rule learners to defect datasets from the PROMISE repository [37]. They indicated that the lessons learned after combining small parts of different data sources (i.e., the clusters) were superior to either generalizations formed over all the data or local lessons formed from particular projects. Nam et al. proposed novel approaches CLA and CLAMI, which can work well for defect prediction on unlabeled datasets in an automated manner without any manual effort [38]. CLA automatically clusters and labels instances, and CLAMI has two additional steps to select features and instances. They compared CLA/CLAMI with three baselines (i.e., supervised learning, threshold-based and expert-based approaches) in terms of precision, recall, F-score and AUC and demonstrated their practicability.

In addition, datasets of many projects are not totally available due to the privacy policy. To address the privacy problem, Peters et al. studied a lot [39–41]. First, they measured the utility of privatized datasets empirically using Random Forests, Naive Bayes and Logistic Regression, through which they showed the usefulness of their proposed privacy algorithm MORPH [39]. MORPH is a data mutator that moves the data a random distance, while not across the class boundaries. In a later work, they improved MORPH by proposing

CLIFF+MORPH to enable effective defect prediction from shared data while preserving privacy [40]. CLIFF is an instance pruner that deletes irrelevant examples. Recently, they again extended MORPH to propose LACE2 [41].

*5.3. Features*

Features, also named as metrics or attributes, are crucial for a successful defect prediction model. Researchers have argued a lot about feature extraction for defect prediction [42–44].

Menzies et al. reported that code metrics are useful for defect prediction [42]. They compared two decision trees (OneR and J48) and naive Bayes, and found naive Bayes is the best. Moser et al. conducted a comparative analysis about the predictive power of two different sets of metrics (i.e., code metrics and process metrics) for defect prediction [43]. They used three common machine learners: logistic regression, Naive Bayes, and decision tree. They showed that process metrics are more efficient defect predictors than code metrics, especially for cost-sensitive classification. Rahman et al. enhanced the conclusion Moser et al. made by analyzing the applicability and efficacy of process and code metrics [44]. They built four prediction models (i.e., Logistic regression, Naive Bayes, J48 decision tree and support vector machine) across 85 releases of 12 large open source projects to address the performance, stability, portability and stasis of different sets of metrics. They strongly suggested the use of process metrics instead of code metrics.

Except for the above two main sets of metrics (code metrics and process metrics), some researchers have tried other kinds of metrics [45,46]. Lee et al. proposed 56 novel micro interaction metrics (MIMs) that leverage developers' interaction information [45]. They investigated three prediction models, i.e., Bayesian Network, J48 decision tree, and logistic regression. Their experimental results showed that MIMs significantly improve defect prediction. Jiang et al. proposed personalized defect prediction, in which they leveraged developer-specific metrics (such as commit hour and cumulative change count) to build a separate prediction model for each developer to predict software defects [46]. They found that the advantage of personalized defect prediction is not bounded to any classification algorithm. Posnett et al. used logistic regression on defect and process data from 18 open source projects to illustrate the risks of modeling at an aggregation level (e.g., packages) in the context of defect prediction [47]. They found that although it is often necessary to study phenomena based on data at aggregated levels of products, teams, or processes, it is also possible that the resulting findings are only actionable at the disaggregated level, such as at the level of files, individual people, or steps of a process. In defect prediction, the studies that predicting defects at the disaggregated level indeed achieve good results [48–50]. Hata et al. conducted fine-grained bug prediction, which is a method-level prediction, on Java software based on historical metrics [48]. They used random forest and found that method-level prediction is more effective than package-level and file-level prediction when considering efforts, which is because predicted buggy packages and files contain many non-buggy packages and files. Kim et al. introduced a new technique for change-level defect prediction [49]. They are the first to classify file changes as buggy or clean leveraging change information features. They used support vector machine to determine whether a new software change is more similar to prior buggy changes or clean changes. A big advantage of change classification is that predictions can be made immediately upon completion of a change. Kamei et al. conducted a large-scale study of six open source and five commercial projects from multiple domains for change-level defect prediction [50]. They used a logistic regression model based on 14 change metrics. They expected that change-level defect prediction can provide an effort-minimizing way to focus on the most risky changes and thus reduce the costs of building high-quality software.

With many different features, researchers also tried to pre-process the features in order to gain higher-quality feature sets [51]. Shivaji et al. investigated multiple feature selection techniques that are generally applicable to classification-based bug prediction methods [51]. They used Naive Bayes and Support Vector Machine. They found that binary

features are better, and between 3.12% and 25% of the total feature set yielded optimal classification results.

Recently, deep learning, as an advanced prediction model, is more and more popular. The role of deep learning is to automatically generate features which are better for prediction model building. Therefore, several researchers have also tried to improve the performance of defect prediction via deep learning [52–54]. Jiang et al. proposed a cost-sensitive discriminative dictionary learning (CDDL) approach for software defect prediction [52]. CDDL is based on sparse coding which can transform the initial features into more representative code. Their results showed that CDDL is superior to five representative methods, i.e., support vector machine, Compressed C4.5 decision tree, weighted Naive Bayes, coding based ensemble learning (CEL), and cost-sensitive boosting neural network. Wang et al. leveraged Deep Belief Network (DBN) to automatically learn semantic features from token vectors extracted from programs' Abstract Syntax Trees [54]. Their evaluation on ten open source projects showed that learned semantic features significantly improve both within-project defect prediction and cross-project defect prediction compared to traditional features.

*5.4. Algorithms*

Lessmann et al. proposed a framework for comparative software defect prediction experiments [1]. They conducted a large-scale empirical comparison of 22 classifiers over 10 public domain data sets from the NASA Metrics Data repository (http://mdp.ivv. nasa.gov, accessed on 29 January 2023) and the PROMISE repository (http://promise.site. uottawa.ca/SERepository, accessed on 29 January 2023). Their results indicated that the importance of the particular classification algorithm may be less than previously assumed since no significant performance differences could be detected among the top 17 classifiers. However, Ghotra et al. doubted the conclusion and they pointed that the datasets Lessmann et al. used were both noisy and biased [55]. Therefore, they replicated the prior study with initial datasets as well as the datasets after cleanse. They found that some classification techniques tend to produce defect prediction models that outperform others. They showed that Logistic Model Tree when combined with ensemble methods (i.e., bagging, random subspace, and rotation forest) achieves top-rank performance. Furthermore, clustering techniques (i.e., Expectation Maximization and K-means), rule-based techniques (Repeated Incremental Pruning to Produce Error Reduction and Ripple Down Rules), and support vector machine are worse.

Many algorithms have several tunable parameters and their values may have an impact on the prediction performance. Tantithamthavorn et al. conducted a case study on 18 datasets to investigate an automated parameter optimization technique, Caret, in defect prediction [56]. They tried 26 classification techniques that require at least one parameter setting and concluded that automated parameter optimization techniques such as Caret yield substantially benefits in terms of performance improvement and stability, while incurring a manageable additional computational cost.

*5.5. Cross-Project Defect Prediction*

Cross-project defect prediction is a rising topic. It uses data from one project to build the prediction model and predicts defects in another project based on the trained model so that it can solve the problem that there is no sufficient amount of data available to train within a project (such as a new project).

Zimmermann studied cross-project defect prediction models on a large scale [57]. Their results indicated that cross-project prediction is a serious challenge. To face the challenge, they identified factors that do influence the success of cross-project predictions. In addition, they derived decision trees that can provide early estimates. Rahman et al. investigated cross-project defect prediction compared with within-project prediction using logistic regression model [58]. They found that in terms of traditional evaluation metrics precision, recall, F-measure and AUC, cross-project performance is significantly worse than within-

project performance. However, in terms of a cost-sensitive evaluation metric AUCEC (Area Under the Cost-Effectiveness Curve), cross-project defect prediction performs surprisingly well and may have a comparable performance to that of the within-project models.

To improve the performance of cross-project defect prediction, researches have tried several techniques [59–62]. Nam et al. proposed a novel transfer defect learning approach, TCA+, by extending a transfer learning approach Transfer Component Analysis (TCA) [59]. TCA+ can provide decision rules to select suitable normalization options for TCA of a given source-target project pair. In a later work, they addressed the limitation that cross-project defect prediction cannot be conducted across projects with heterogeneous metric sets by proposing a heterogeneous defect prediction approach [61]. The approach conducts metric selection and metric matching to build a prediction model. Jiang et al. proposed an approach CCA+ for heterogeneous cross-company defect prediction [60]. CCA+ combines unified metric representation and canonical correlation analysis and can achieve the best prediction results with the nearest neighbor classifier. Zhang et al. found that connectivity-based unsupervised classifiers (via spectral clustering) offer a viable solution for cross-project defect prediction [62]. Their spectral classifier ranks as one of the top classifiers among five widely-used supervised classifiers (i.e., random forest, naive Bayes, logistic regression, decision tree, and logistic model tree) and five unsupervised classifiers (i.e., k-means, partition around medoids, fuzzy C-means, neural-gas, and spectral clustering) in cross-project defect prediction.

## 6. Software Management

Software management aims to manage the whole lifetime of software. It includes software requirement engineering, software development process, evaluate software cycle time and forecast software evolution, in order to improve software quality and development efficiency.

### 6.1. Software Requirement Engineering

Yang et al. used machine learning algorithms to determine whether an ambiguous sentence is nocuous or innocuous, based on a set of heuristics that draw on human judgments [63]. They focused on coordination ambiguity. They found that LogitBoost algorithm performed better than other candidates including decision trees, J48, Naive Bayes, SVM, and Logistic Regression. Anish et al. conducted a study to identify, document, and organize Probing Questions (PQs) for five different areas of functionality into structured flows [64]. They used Naive Bayes to identify Architecturally Significant Functional Requirements (ASFRs), used random k labelsets classifier to categorize ASFR by types, and finally recommended PQ-flows.

### 6.2. Software Development Process

To manage software development process, Chen et al. presented a novel semi-automated approach to software process evaluation using machine learning techniques [65]. They formulated the problem as a sequence classification task and defined a new quantitative indicator named process execution qualification rate to objectively evaluate the quality and performance of a software process. They used decision tree, Naive Bayes (NB) classifier, and Support Vector Machine (SVM), and found SVM achieves the best performance. Blincoe et al. investigated what work dependencies should be considered when establishing coordination needs within a development team [66]. They used their conceptualization of work dependencies, named Proximity, and leveraged k-nearest neighbor machine learning algorithm to analyze what additional task properties are indicative of coordination needs, which are defined as those that can cause the most disruption to task duration when left unmanaged.

### 6.3. Software Cycle Time

To evaluate software finish time, Nan et al. formalized the nonlinear effect of management pressures on project performance as U-shaped relationships using Regression Models [67]. They found that controlling for software process, size, complexity, and conformance quality, budget pressure, a less researched construct, has significant U-shaped relationships with development cycle time and development effort. Choetkiertikul et al. proposed a novel approach to provide automated support for project managers and other decision makers in predicting whether a subset of software tasks in a software project have a risk of being delayed [68]. They used both local data and networked data, and use random forest as the local classifier. In addition, they use d collective classification to simultaneously predict the degree of delay for a group of related tasks.

### 6.4. Software Evolution

Chaikalis et al. attempted to model several aspects of graphs representing object-oriented software systems as they evolve over a number of versions [69]. They developed a prediction model by considering global phenomena such as preferential attachment, past evolutionary trends such as the tendency of classes to create fewer relations as they age, as well as domain knowledge in terms of principles that have to be followed in object-oriented design. The derived models can provide insight into the future trends of software and potentially form the basis for eliciting improved or novel laws of software evolution.

## 7. Software Quality

In this section, we mainly introduce three sub tasks in software quality tasks, which are software reliability prediction, software vulnerability prediction and malware detection.

### 7.1. Software Reliability Prediction

To measure software quality, many studies have been conducted to predict software reliability [70–73].

Wilson et al. presented a nonparametric software reliability model based on the order-statistic paradigm [70]. The approach makes use of Bayesian nonparametric inference methods and consciously eschewed the "random sampling" assumption. Cheung et al. developed a software component reliability prediction framework [71]. They exploited architectural models and associated analysis techniques, stochastic modeling approaches, and information sources available early in the development lifecycle. They illustrated the utility of their framework as an early reliability prediction approach. Torrado et al. described statistical inference and prediction for software reliability models in the presence of covariate information [72]. They developed a semiparametric, Bayesian model using Gaussian processes to estimate the numbers of software failures over various time periods when it is assumed that the software is changed after each time period and that software metrics information is available after each update. Misirli et al. investigated the applications of Bayesian networks in software engineering in terms of topics, structure learning, parameter learning, and variable types [73]. They proposed a Hybrid Bayesian network, which utilizes techniques that are widely used in other disciplines, such as dependence analysis for structure learning and Gibbs sampling for inference, for software reliability prediction.

There are also other studies that focus on web service reliability [74–76]. Zheng et al. proposed a collaborative reliability prediction approach [74], which employs the past failure data of other similar users to predict the web service reliability for the current user, without requiring real-world web service invocations. In a later work, Zheng et al. proposed two personalized reliability prediction approaches for web services, that is, neighborhood-based approach and model-based approach [75]. The neighborhood-based approach employs past failure data of similar neighbors (either service users or web services) to predict the web service reliability. The model-based approach fits a factor model based on the available web service failure data and uses this factor model to make further

reliability prediction. Silic et al. presented CLUS, a model for reliability prediction of atomic web services [76]. They used k-means clustering to improve the accuracy of the current state-of-the-art prediction models by considering user–, service– and environment–specific parameters of the invocation context.

### 7.2. Software Vulnerability Prediction

The other aspect of software quality is software vulnerability, which is also a key measure for software quality. There are several studies about vulnerability prediction [77–79].

Shin et al. investigated whether software metrics obtained from source code and development history are discriminative and predictive of vulnerable code locations [77]. They used three categories of metrics: complexity, code churn, and developer activity. They tried Logistic regression, J48 decision tree, Random forest, Naive Bayes, and Bayesian network, among which Naive Bayes performs the best. Shar et al. proposed the use of dynamic attributes to complement static attributes in vulnerability prediction [78]. They first applied min-max normalization and then principle component analysis to every dataset collected. They used a subset of principal components as attributes such that it explained at least 95% of the data variance. They compared Logistic Regression (LR) and Multi-Layer Perceptron (MLP) and advised the use of LR. They also used k-means clustering for unsupervised vulnerability prediction. Scandariato et al. presented a machine learning approach to predict which components of a software application contain security vulnerabilities [79]. The approach is based on text mining the source code of the components. They investigated 20 Android applications and explored five well-known learning techniques: Decision Trees, k-Nearest Neighbor, Naive Bayes, Random Forest and Support Vector Machine. The best two are Naive Bayes and Random Forest.

### 7.3. Malware Detection

Some software, especially mobile apps, can be malware. They implement functionalities which contradict with user interests. Generally, malwares, which include viruses, worms, trojans and spyware, are harmful at diverse severity scales. Malware can lead to damages of varying severity, ranging from spurious app crashes to financial losses with malware sending premium-rate SMS, as well as to private data leaks. There are many studies which are aimed to detect malwares [80,81].

Chandramohan et al. proposed and evaluated a bounded feature space behavior modeling (BOFM) framework for scalable malware detection. They first extracted a feature vector which is bounded by an upper limit N using BOFM, and then they tried support vector machine and logistic regression and found support vector machine is better for malware detection [80]. Avdiienko et al. used data flow of apps to detect malwares [81]. They first used weighted Jaccard distance metric to compute an app to its k-nearest neighbors for each source category of the app, in which they determined weight by the mean values of all apps within a source category. Then they formed a vector and used v-SVM one-class classifier to detect malwares.

## 8. Software Performance Prediction

Among the software performance prediction models, there are mainly two categories of models found in literature, i.e., white-box models and black-box models. White-box models are built early in the life cycle, by studying the underlying design and architecture of the software system in question. White-box models include Queueing networks, Petri Nets, Stochastic Process Algebras, and etc. On the contrary, black-box models do not make any assumption on the design and architecture, but effectively treating the system as a black box.

### 8.1. White-Box Models

White-box models can identify performance bottlenecks early, so that developers can take corrective actions.

Jin et al. introduced a systematic approach BMM to evaluate and predict the performances of database-centric information systems when they are subject to an exorbitant growth of workload [82]. BMM combines benchmarking, production system monitoring, and performance modelling. They used layered queueing network (LQN) as the performance modelling method. Krishnamurthy et al. introduced the Weighted Average Method (WAM) to improve the accuracy of analytic predictive performance models for systems with bursts of concurrent customers [83]. It is more robust with respect to the distributions that contribute to bursty behaviour and it improves both queuing network models (QNM) and layered queueing models (LQM). Rathfelder et al. presented an automated performance prediction approach in the context of capacity planning for event-based systems [84]. The approach is based on the Palladio Component Model (PCM), which is a performance meta-model for component-based systems. Koziolek et al. applied a novel, model-driven prediction method called Q-ImPrESS on a large-scale process control system consisting of several million lines of code from the automation domain to evaluate its evolution scenarios [85]. For performance prediction, the Q-ImPrESS workbench creates instances of the Palladio Component Model (PCM). Internally, it can solve a model either using simulation or using numerical analysis based on an additional transformation into layered queueing networks.

### 8.2. Black-Box Models

In general, black-box models are grouped into two categories, i.e., model-based techniques and measurement-based techniques.

For the model-based techniques, Guo et al. proposed a variability-aware approach to performance prediction via statistical learning [86]. They used decision tree CART to predict performance of configurable systems based on randomly selected configurations. Sarkar et al. used a combination of random sampling and feature-coverage heuristics to dynamically build the initial sample [87]. In particular, they proposed a feature-frequency heuristic for the initial sample generation. They build prediction models using Classification and Regression Tree (CART), to demonstrate the superiority of their sampling approach, i.e., projective sampling using the feature-frequency heuristic. Zhang et al. proposed a novel algorithm based on Fourier transform for performance prediction [88]. The algorithm is able to make predictions of any configurable software system with theoretical guarantees of accuracy and confidence level specified by the user, while using minimum number of samples up to a constant factor.

For the measurement-based techniques, Westermann et al. proposed an automated, measurement-based model inference method to derive goal-oriented performance prediction functions [89]. They used adaptive selection of measurement points and investigated four models, i.e., Multivariate Adaptive Regression Splines, Classification and Regression Trees, Genetic Programming and Kriging. Siegmund et al. presented a method that automatically detects performance-relevant feature interactions to improve prediction accuracy [90]. They proposed three heuristics to reduce the number of measurements required to detect interactions. In a later work, they proposed an approach that derives a performance-influence model for a given configurable system, describing all relevant influences of configuration options and their interactions [91]. Their approach combines machine-learning and sampling heuristics in a novel way.

### 8.3. Performance-Related Analysis

In addition to the above models for performance prediction, there are other studies about performance-related analysis [92,93].

Acharya et al. presented a novel framework called PerfAnalyzer, a storage-efficient and pro-active performance monitoring framework for correlating service health with performance counters [92]. PerfAnalyzer uses three ML algorithms, i.e., Decision Tree (DT), Naive Bayes (NB), and Dichotomous Logistic Regression (DLR) to produce health models. Malik et al. presented and evaluated one supervised and three unsupervised approaches

for performance analysis [93]. They found that their wrapper-based supervised approach, which uses a search-based technique to find the best subset of performance counters and a logistic regression model for deviation prediction, can provide up to 89% reduction in the set of performance counters while much accurately detecting performance deviations.

## 9. Effort Estimation

In this section, we mainly introduce two sub tasks in effort estimation tasks, which are software effort estimation and web effort estimation.

### 9.1. Software Effort Estimation

Software effort estimation is an important activity in the software development process. There are many studies related to effort estimation. However, in the journals and proceedings we investigate, the related papers are not in a great amount.

In 2007, Jorgensen et al. conducted a systematic review of previous work about software effort estimation, which provide a basis for the improvement of software estimation research through [94]. They identified 304 software cost estimation papers in 76 journals and classified the papers according to research topic, estimation approach, research approach, study context and data set. The estimation approaches include regression, decision tree, neural network and Bayesian methods.

Subsequent studies proposed several more advanced techniques for software effort estimation [95,96]. Kultur et al. proposed to use ensemble of neural networks to estimate software effort [95]. They first used bootstrapping to train several multilayer perceptrons, and then used ART algorithm to find the largest cluster and ensembled the results of the multilayer perceptrons in the cluster. Whigham et al. proposed an automatically transformed linear model (ATLM) as a suitable baseline model for comparison against software effort estimation methods [96]. ATLM is simple yet performs well over a range of different project types. In addition, ATLM may be used with mixed numeric and categorical data and requires no parameter tuning.

In addition, some researchers conducted comparative studies to investigate which estimation approach performs the best for software effort estimation [97,98]. Dejaeger et al. conducted a comparative study of data mining techniques for software effort estimation [97]. The techniques include tree/rule-based models such as M5 and CART, linear models such as various types of linear regression, nonlinear models (MARS, multilayered perceptron neural networks, radial basis function networks, and least squares support vector machines), and estimation techniques that do not explicitly induce a model (e.g., a case-based reasoning approach). The results showed that least squares regression in combination with a logarithmic transformation performs best. Mittas et al. proposed a statistical framework based on a multiple comparisons algorithm in order to rank several cost estimation models, identifying those which have significant differences in accuracy and clustering them in non-overlapping groups [98]. They examined the predictive power of 11 models over 6 public domain datasets. The results showed that very often a linear model is adequate enough, but there is not a global solution.

### 9.2. Web Effort Estimation

Besides software effort estimation, there are several studies about other kinds of effort estimation such as web effort estimation [99].

Mendes et al. compare, using a cross-company data set, eight Bayesian Network (BN) models for web effort estimation as well as Manual Stepwise Regression (MSWR), Case-Based Reasoning (CBR), and mean and median-based effort models [99]. MSWR presented significantly better predictions than any of the BN models built herein and, in addition, was the only technique to provide significantly superior predictions to a median-based effort model. They suggest that the use of simpler models, such as the median effort, can outperform more complex models, such as BNs. In addition, MSWR seemed to be the only effective technique for web effort estimation.

## 10. Software Testing

Software testing is very important in the software development process. The better the testing is, the fewer bugs exist.

### 10.1. Test Case Quality

In software testing, test case quality is the most important factor. Good test cases can effectively improve the software quality. On the contrary, bad test cases may cost much time and effort while still not find the defects or faults. There are many studies that are aimed to improve the test case quality and reduce test effort [100–104].

To guarantee test case quality, Natella et al. proposed a new approach to refine the faultload by removing faults that are not representative of residual software faults [100]. They used decision tree to classify whether a fault is representative or not based on software complexity metrics (i.e., the number of statements and the number of paths in a component, and the number of connections between components). Their approach can be used for improving fault representativeness of existing software fault injection approaches. Cotroneo et al. presented a method based on machine learning to combine testing techniques adaptively during the testing process [101]. The method contains an offline learning phase and an online learning phase. In offline learning, they first defined the features of a testing session potentially related to the techniques performance, and then used several machine learning approaches (i.e., Decision Trees, Bayesian Network, Naive Bayes, Logistic Regression) to predict the performance of a testing technique. In online learning, they adapt the selection of test cases to the data observed as the testing proceeds. Yu et al. proposed a technique that can be used to distinguish failing tests that executed a single fault from those that executed multiple faults [102]. The technique suitably combines information from a set of fault localization ranked lists, each produced for a certain failing test, and the distance between a failing test and the passing test that most resembles it.

To reduce test effort, Song et al. proposed a new algorithm called interaction tree discovery (iTree) that aims to identify sets of configurations to test [103]. The sets of configurations are smaller than those generated by CIT, while also including important high-strength interactions missed by practical applications of CIT. They first used a fast and effective clustering algorithm CLOPE to cluster configurations and then used decision tree to discover commonalities among configurations in each of the clusters. In a later work, they presented an improved iTree algorithm in greater detail [104]. The key improvements are based on the use of composite proto-interactions, i.e., a construct that improves iTree's ability to correctly learn key configuration option combinations, which in turn significantly improves iTree's running time, without sacrificing effectiveness.

### 10.2. Test Application

With many good test cases, various tasks can be carried out to ensure the software quality [105–107].

Ali et al. undertook the task of preparing a new subject program for use in fault localization experiments, one that has naturally-occurring faults and a large pool of test cases written by impartial human testers [105]. They investigated the performances of five standard techniques (ConjunctiveRule, JRip, OneR, PART, and Ridor) from the data mining tool Weka in fault localization. They found that standard classifiers suffer from the class imbalance problem. However, they found that adding cost information improves accuracy. Farzan et al. proposed null-pointer dereferences as a target for finding bugs in concurrent programs using testing [106]. A null-pointer dereference prediction engine observes an execution of a concurrent program under test and predicts alternate interleavings that are likely to cause null-pointer dereferences. They used an abstraction to the shared-communication level, took advantage of a static lock-set based pruning, and finally, employed precise and relaxed constraint solving techniques that use an SMT solver to predict schedules. Nori et al. described an algorithm TpT for proving termination of a program based on information derived from testing it [107]. In TpT, linear regression is

used to efficiently compute a candidate loop bound for every loop in the program. If all loop bounds are valid, then there is a proof of termination.

## 11. Program Analysis

There are many studies that analyse program execution data for various targets, which can be grouped mainly into two categories, i.e., dynamic analysis and static analysis.

### 11.1. Dynamic Analysis

Dynamic program analysis, which analyses a program during its execution, is a common technique. When a program is executed, it has many medium results and outputs. With these data, many kinds of analyses can be performed [108–111].

Haran et al. presented and studied three techniques (i.e., random forests, basic association trees, and adaptive sampling association trees) to automatically classify program execution data as coming from program executions with specific outcomes (i.e., passing or failing) [108]. The techniques first build behavioral models by feeding suitably labeled execution data to statistical learning algorithms, and then use the behavioral models to classify new (unknown) executions. Yilmaz et al. presented a hybrid instrumentation approach which combines hardware and software instrumentation to classify program executions based on decision tree [109]. They used hardware performance counters to gather program spectra at very low cost. These underlying data are further augmented with data captured by minimal amounts of software-level instrumentation. Xiao et al. proposed a fully automated approach TzuYu to learn stateful typestates (which are important for program debugging and verification) from Java programs [110]. The approach extends the classic active learning process to generate transition guards (i.e., propositions on data states) Specifically, the approach combined the active learning algorithm L* with a random argument generation technique, and then used support vector machine to abstract data into propositions. Lee et al. presented a machine learning-based framework for memory leak detection [111]. The framework observes the staleness of objects during a representative run of an application. From the observed data, the framework generates training examples, which also contain instances of hypothetical leaks. Based on the training examples, the proposed framework leverages support vector machine to detect memory leak.

### 11.2. Static Analysis

In addition to dynamic analysis, program can be analysed statically as well [112–114]. Compared to dynamic analysis, static analysis costs less effort.

Bodden et al. presented ahead-of-time techniques that can prove the absence of property violations on all program runs, or flag locations where violations are likely to occur [112]. They focused on tracematches, an expressive runtime monitoring notation for reasoning about groups of correlated objects. They described a novel flow-sensitive static analysis for analyzing monitor states, in which they proposed a machine learning phase to filter out likely false positives. That is, they used decision tree to classify points of failure based on seven features, each of which is present at a point of failure if the static analysis encounters the feature on the point of failure itself or on a "context shadow". Tripp et al. presented a novel approach for adaptive program parallelization [113]. The approach permits low-overhead, input-centric runtime adaptation by shifting most of the cost of predicting per-input parallelism to an expensive offline analysis. They used linear regression to seek correlations between input features and parallelism levels. Sun et al. presented a novel technique IntEQ to recognize benign IOs via equivalence checking across multiple precisions [114]. They determined if an IO is benign by comparing the effects of an overflowed integer arithmetic operation in the actual world (with limited precision) and the same operation in the ideal world (with sufficient precision to evade the IO).

## 12. Traceability

Software traceability is an important element of the development process, especially in large, complex, or safety critical software-intensive systems. It is used to capture relationships between various software development artifacts (such as requirements documents, design documents, code, bug reports, and test cases), and support critical activities such as impact analysis, compliance verification, test-regression selection, and safety-analysis.

### 12.1. Software Traceability

There are various studies about software traceability [115–117].

Asuncion et al. proposed an automated technique that combines traceability with a machine learning technique known as topic modeling [115]. They automatically records traceability links during the software development process and learns a probabilistic topic model over artifacts. Wu et al. developed an automatic link recovery algorithm, ReLink, which automatically learns criteria of features from explicit links to recover missing links [116]. They used decision tree to build classification model based on the identification of features of the links between bugs and changes. They showed that ReLink is better than traditional heuristics in software maintainability measurement and defect prediction. Nguyen et al. introduced MLink, a multi-layered approach that takes into account not only textual features but also source code features of the changed code corresponding to the commit logs [117]. MLink combined patch-based, name-based, text-based and association-based detector as well as a filtering layer. MLink is also capable of learning the association relations between the terms in bug reports and the names of entities/components in the changed source code of the commits from the established bug-to-fix links, and uses them for link recovery between the reports and commits that do not share much similar texts.

In addition to the above traditional software traceability tasks, there are other novel ones [118–120].

Grechanik et al. proposed a novel approach for automating part of the process of recovering traceability links between types and variables in Java programs and elements of use case diagrams [118]. They used Naive Bayes classifier (compute the probability) to recover traceability links. The results suggested that our approach can recover many TLs with a high degree of automation and precision. Mirakhorli et al. presented a novel approach for automating the construction of traceability links for architectural tactics [119]. They investigated five tactics and utilized machine learning methods and lightweight structural analysis to detect tactic-related classes. Specifically, they utilized an algorithm that they had previously developed to detect non-functional requirements (NFRs). The algorithm matched or outperformed standard classification techniques including the naive bayes classifier, standard decision tree algorithm (J48), feature subset selection (FSS), correlation based feature subset selection (CFS), and various combinations of the above for the specific task of classifying NFRs. In a later work, Mirakhorli et al. went further by investigating 10 tactics [120]. They used the same algorithm above for discovering and visualizing architectural tactics in code, mapping these code segments to tactic traceability patterns, and monitoring sensitive areas of the code for modification events in order to provide users with up-to-date information about underlying architectural concerns. They compared the performance of their algorithm with six off-the-shelf classifiers (support vector machine (SVM), C.45 decision tree (DT) (implemented as J48 in Weka), Bayesian logistic regression (BLR), AdaBoost, rule learning with SLIPPER, and bagging).

### 12.2. Traceability Quality

To guarantee the traceability quality, Lohar et al. presented a novel approach to trace retrieval in which the underlying infrastructure is configured at runtime to optimize trace quality [121]. They used Genetic Algorithm to search for the best configuration given an initial training set of validated trace links, a set of available tracing techniques specified in a feature model, and an architecture capable of instantiating all valid configurations of features.

## 13. Bug Report Management

Bugs are inevitable in the software development process. When a bug is found, a corresponding bug report will be generated. To better handle so many software bugs, bug reports should be managed carefully.

### 13.1. Bug Report Quality

In bug report management, the first concern is the quality of bug reports. Good bug reports can help developers fix the bugs efficiently, while bad bug reports may waste developers much time and effort.

Bettenburg et al. investigated what makes a good bug report [122]. They measured the quality of a bug report based on seven features (i.e., itemizations, keyword completeness, code samples, stack traces, patches, screenshots and readability), and built three supervised learning models (i.e., support vector machine, generalized linear regression, and stepwise linear regression). Their results suggested that steps to reproduce and stack traces are most useful in bug reports. Zanetti et al. proposed an efficient and practical method to identify valid bug reports (i.e., the bug reports that refer to an actual software bug, are not duplicates and contain enough information to be processed right away) [123]. They used support vector machine to identify valid bug reports based on nine network measures using a comprehensive data set of more than 700,000 bug reports obtained from the BUGZILLA installation of four major OSS communities.

To reduce redundant effort, there are many researches about identification of duplicate bug reports [124–128]. Runeson et al. proposed to use Natural Language Processing techniques to support the duplicate bug report identification [124]. They only used cosine similarity to rank and identify duplicates. Wang et al. leveraged additional execution information [125]. Although their additional features improve the performance of duplicate bug report identification, their algorithm is the same as Runeson et al.'s (i.e., cosine similarity). Sun et al. used discriminative models to identify duplicates more accurately [126]. They used support vector machine with linear kernel based on 54 text features. In a later work, they proposed a retrieval function REP to measure the similarity between two bug reports [127]. They fully utilized the information available in a bug report including not only the similarity of textual content in summary and description fields, but also similarity of non-textual fields such as product, component, version, etc, and they used a two-round stochastic gradient descent to automatically optimize REP in a supervised learning manner. Liu et al. leveraged ranking SVM, a Learning to Rank technique to construct a ranking model for effective (i.e., non-duplicate) bug report search [128]. They used textual, semantic and other categorical and numerical features.

To effectively analyze anomaly bug reports, Lucia et al. proposed an approach to automatically refine bug reports by the incorporation of user feedback [129]. They first presented top few anomaly reports from the list of reports generated by a tool in its default ordering. Users then either accepted or rejected each of the reports. Based on the feedback, their system automatically and iteratively refined a classification model for anomalies and resorted the rest of the reports. They used a variant of nearest neighbor classification scheme, namely nearest neighbor with non-nested generalization (NNGe).

### 13.2. Bug Report Assignment and Categorization

For the bug reports that are valid, how to make a proper and effective assignment is an important task since it may affect the efficiency of development process. Proper bug report assignment can save much time and effort on bug-fixing activity.

Anvik et al. presented a semi-automated approach intended to ease the assignment of reports to a developer [130]. They used the one-line summary and full text description to characterize each bug report. They chose support vector machine as the final algorithm after comparing it with Naive Bayes and C4.5 decision tree. In a later work, Anvik et al. presented a machine learning approach to create recommenders that assist with a variety of decisions aimed at streamlining the development process [3]. They used the approach to create three

different kinds of development-oriented recommenders: a developer recommender that suggests which developers might fix a report, a component recommender that suggests to which product component a report might pertain, and an interest recommender that suggests which developers on the project might be interested in following the report. For the developer recommender, they investigated Naive Bayes, Support Vector Machines, C4.5 decision tree, Expectation Maximization, conjunctive rules, and nearest neighbour. Jeong et al. introduced a graph model based on Markov chains, which captures bug tossing history [131]. They showed that the accuracy of bug assignment prediction is improved using naive Bayes with tossing graph.

Proper bug report assignment also relies on proper bug report categorization. There are a number of studies that propose techniques to categorize bug reports. Among them a popular research area is reopened bug prediction [132,133].

Zimmermann et al. characterized and predicted which bugs are reopened [132]. They first qualitatively identified causes for bug reopens, and then built logistic regression model to predict the probability that a bug will be reopened. Xia et al. proposed a novel approach ReopenPredictor which extract more textual features from the bug reports [133]. ReopenPredictor combines decision tree and multinomial Naive Bayes to yield better performance in reopened bug prediction.

In addition, Xuan et al. leveraged the developer prioritization to improve three predicted tasks in bug repositories, i.e., bug triage, severity identification, and reopened bug prediction [134]. For bug triage, they used SVM and NB. For severity identification, they used NB. For reopened bug prediction, they used Adaboost.

### 13.3. Bug Fix Related Task

Based on bug reports, there are also many other studies which are related to bug fixes [135–137].

Kim et al. proposed a two-phase prediction model that uses bug reports' contents to suggest the files likely to be fixed [135]. In the first phase, they checked whether the given bug report contains sufficient information for prediction (predictable or deficient). In the second phase, they proceeded to predict files to be fixed for the predictable bug report. Zhang et al. proposed a Markov-based method for predicting the number of bugs that will be fixed in future [136]. For a given bug report, they also constructed a KNN classification model to predict slow or quick fix (e.g., below or above a time threshold), which is based on the assumption that the similar bugs could require similar bug-fixing effort. The features they used are submitter, owner, severity, ESC (which indicates whether the bug is an externally discovered bug reported by end users or an internally discovered bug reported by the QA team), priority, category and summary. Guo et al. performed an empirical study to characterize factors that affect which bugs are fixed in Windows Vista and Windows 7 [137]. They focused on factors related to bug report edits and relationships between people involved in handling the bug, and built a statistical model using logistic regression to predict the probability that a new bug will be fixed.

## 14. Developers and Users

Developers and users play the key role in the software development process. Researching on the developer-related and user-related data has a great value [138–147].

### 14.1. Developer Related Task

Meneely et al. examined the structure of developer collaboration with the developer network derived from code churn information to predict software failures at the file level [138]. They tried three generalized linear regressions previously used for predicting failure count data, i.e., negative binomial regression, Poisson regression, and logistic regression, and they used logistic regression as the final model. Later, Meneely et al. performed an empirical and longitudinal case study of a large Cisco networking product over a five year history [139]. They examined statistical correlations between monthly

team-level metrics and monthly product-level metrics. Their linear regression prediction model based on team metrics was able to predict the product's post-release failure rate within a 95% prediction interval for 38 out of 40 months.

Müller et al. investigated developers' emotions, progress and the use of biometric measures to classify them in the context of software change tasks [142]. They used J48 decision tree to distinguish between positive and negative emotions based on biometric measurements, i.e., electro-dermal activity, electroencephalography, skin temperature, heart rate, blood volume pulse and various eye-related measurements, such as pupil size. Later, Müller et al. investigated the use of biometrics to determine code quality concerns with ten professional developers [143]. They used random forest and their results showed that biometrics are indeed able to predict quality concerns of parts of the code while a developer is working on, improving upon a naive classifier by more than 26% and outperforming classifiers based on traditional metrics.

Bacchelli et al. presented an approach to classify email content at line level [144]. Their technique fuses an automated supervised machine learning approach (i.e., naive Bayes) with island parsing to perform automatic classification of the content of development emails into five language categories: natural language text, source code fragments, stack traces, code patches, and junk. Their technique can help one to subsequently apply ad hoc analysis techniques for each category. Later, Sorbo et al. proposed a semi-supervised approach named DECA (Development Emails Content Analyzer) to mine intention from developer emails [145]. DECA uses Natural Language Parsing to classify the content of development emails according to their purpose (e.g., feature request, opinion asking, problem discovery, solution proposal, and information giving), identifying email elements that can be used for specific tasks. They showed the superiority of DECA to traditional machine learning techniques (i.e., naive Bayes classifier, the Logistic Regression, Simple Logistic, J48, the alternating decision tree (ADTree), Random Forest, FT, Ninge).

*14.2. User Related Task*

Murukannaiah et al. proposed the Platys framework as a way to address the special challenges of place-aware application development [147]. They collected place labels and Android phone sensor readings from 10 users, and applied Platys to learn each user's places. Platys combines active learning and semi-supervised learning. In active learning, it prompts the user to label the place for an instance predicted with the least confidence. In semi-supervised learning, it uses the predicted label with the biggest confidence.

## 15. Discussion

Based on the reviewed papers, we propose several research questions (RQ) and conclude several challenges and directions from this taxonomy.

*15.1. RQ1: What Features Are Appropriate to Build Prediction Models in Software Engineering Researches?*

**Challenge: good feature extractor**

For various research topics, researchers extract many different features. Although many features are extracted by experts, not all of them are good for building prediction models. First, the features may be correlative or dependent, which are not proper for build some prediction models. Second, the number of features may be too much, and even more than the number of training instances, which will cause overfitting in building prediction models. Therefore, feature extraction becomes a big challenge.

A good feature extractor should meet two main conditions. First, it should be automatic so that it will not cost too much manual effort. Second, it can reduce feature dimension while keep feature quality. That is, given a specific task, the feature extractor can identify and preserve relevant features and remove irrelevant features to improve the performance of prediction models.

**Direction: deep learning**

Studies that apply prediction models to software engineering tasks have last for over ten years. Although researchers have achieved significant improvement from simple linear regression to complex ensemble learning, they have been close to bottlenecks for many software engineering tasks. Recently, deep learning, as an advanced prediction model, has been more and more popular. Some studies that tried to leverage deep learning to software engineering tasks have achieved even better performance than start-of-the-art techniques [52–54]. However, till now the application of deep learning to software engineering tasks is still limited in number.

Actually, the biggest advantage of deep learning is that it can automatically generate more expressive features that are better for learning prediction models, which cannot be accomplished by traditional prediction models. As mentioned above, many software engineering tasks face the feature extraction challenge, i.e., either it is hard to manually infer proper features or there are too much features needing to carefully selected for learning prediction models. Therefore, leveraging deep learning to various software engineering tasks to improve their performances is a promising direction. In 2015, we tried deep learning in just-in-time defect prediction and have achieved better performance than start-of-the-art techniques [53]. In the paper, we generate a set of more expressive features from a set of initial features by leveraging a deep belief network.

### 15.2. RQ2: What Datasets Are Appropriate to Build Prediction Models in Software Engineering Researches?

**Challenge: dataset quality and scale**

A good prediction model heavily relies on the datasets it learns from. With various research topics, there are a large number of datasets that can be used for experiments. However, the datasets vary much in quality (such as bias, noise, size and imbalance), which will influence the effectiveness of prediction models. For example, a dataset with much noise may cause underfitting in building prediction models, a dataset with small size may cause overfitting in building prediction models, and a heavily imbalanced dataset may even fail to build prediction models.

The above problem lead to three major challenges. First of all, there should be formal indicators to quantitively evaluate the quality of datasets. The indicators should consider both the bias, noise, imbalance, size of datasets and the characteristics of the corresponding tasks. Second, based on those indicators, we should pick up some benchmark datasets for each of the research topics. These datasets should be well preprocessed and representative so that various prediction models can be fairly compared based on them. Last but not least, for the datasets that have bad quality but need to be investigated, we should have a step–by–step process to preprocess them so that they can reach the acceptable quality.

In addition, datasets of many projects are not totally available due to the privacy policy, which may influence the generality of the study results. Building effective prediction models from shared data while preserving privacy is also a big challenge.

**Direction: big data and cross-project prediction**

The above challenges and problems lead to big data. That is, studies tend to be large-scale. The biggest advantage of big data is that it generally leads to robust prediction models. Many software engineering tasks value on practicability and generality. If a study only use several small datasets to build prediction models, the results are more likely to have occasionality and the generated prediction models cannot be generalized and are not practical. On the contrary, results achieved from big data are more convincing and more likely to have generality. In addition, deep learning mentioned above also require a large amount of data to learn prediction models, since in deep learning there are much more parameters to learn than traditional prediction models. Note that with the trend of big data, reducing the time and space cost for data computing can also become challenging.

In addition, we can see that many software engineering tasks make analysis and prediction based on project data from the taxonomy. Most of them are in a within-project setting, in which they build prediction models based on historical data in the same project.

However, with the rapid development of software engineering, many new projects are built everyday. When a new project needs to be analyzed, there is no sufficient amount of data available to train within the project. Cross-project prediction can well solve the problem. There have been several studies about cross-project defect prediction [59–62]. For many other research topics, cross-project prediction may show its superiority and attract more and more attention.

### 15.3. RQ3: What Prediction Models Are Appropriate in Different Software Engineering Researches?

**Challenge: good prediction models**

From the taxonomy, we can see that there are various software engineering tasks leveraging prediction models. However, different prediction models fit well in different tasks. With various tasks and various prediction models, there is a need to have reliable guidelines for how to select a good prediction model for a specific task.

**Direction: theoretical guidelines**

Among the reviewed papers, some studies have proposed frameworks for specific tasks to make comparison of various prediction models [1,55,97–99]. However, almost all of them are based on experiments, which always has threats to validity. There should be theoretical guidelines that can provide how to select proper prediction models according to the characteristic of the specific task.

### 15.4. Threats to Validity

Threats to internal validity relate to errors in our taxonomy. We have double checked our search strategy implementation. Still, there could be related papers that we did not notice.

Threats to external validity relate to the generalizability of our conclusions. We have selected three top conference proceedings and two top journals, which can be representative for the state-of-the-art software engineering researches.

In the future, we plan to reduce these threats further by investigating more papers.

## 16. Conclusions

In the paper, we conducted a comprehensive taxonomy on prediction models applied in software engineering. With our search strategy (paper collection, automated filtering and manual filtering), we selected 136 papers from three top conference proceedings (i.e., ICSE, FSE and ASE) and two top journals (i.e., TSE and TOSEM) in the recent 10 years in total.

We found that prediction models have been applied to various software engineering tasks. Based on the software development process, we grouped all tasks into 11 main research topics, i.e., software coding aid, software defect prediction, software management, software quality, software performance prediction, software effort estimation, software testing, software program analysis, software traceability, software bug report management, software users and developers. All the topics play key roles in the software development process. By leveraging prediction models, researchers have achieved good performance in the tasks in these 11 research topics.

Based on the papers, we concluded several big challenges when applying prediction models to software engineering tasks. To achieve better performance in the tasks, we need better datasets, features and methods to select the best prediction models. Researchers shall pay much attentions to these challenges in their later work.

We also showed several promising research directions, some of which are derived by the above challenges. First of all, we can try more deep learning algorithms in the software engineering tasks, since deep learning can extract better features and have shown big potential in many research areas. Second, since deep learning needs large-scale data to train prediction models to achieve better performance, we need to collect more data in every research. In addition, we can try to build more cross-project prediction models to

exploit more data in different projects. We believe that these directions can lead researchers to achieve great improvement in software engineering tasks.

**Author Contributions:** Conceptualization, X.Y. and D.Z.; methodology, X.Y.; investigation, X.Y., J.L. and D.Z.; formal analysis, X.Y.; writing—original draft preparation, X.Y.; writing—review and editing, J.L. and D.Z.; supervision, D.Z.; project administration, D.Z.; funding acquisition, X.Y., J.L. and D.Z. All authors have read and agreed to the published version of the manuscript.

**Funding:** This research is funded by the Natural Science Fundation of Zhejiang Province, China (Grant No.LQ21F020025) and the Basic Public Welfare Research Project of Zhejiang Province (Grant No. LGG21F020008 and Grant No. LGF21F020024).

**Institutional Review Board Statement:** Not applicable.

**Informed Consent Statement:** Not applicable.

**Data Availability Statement:** Not applicable.

**Conflicts of Interest:** The authors declare no conflict of interest.

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
