# Peer review of "A Comprehensive Taxonomy for Prediction Models in Software Engineering"

_information, doi:10.3390/info14020111_

Round 1

Reviewer 1 Report

Authors have made an formal SLR on an interesting topic: Prediction Models in Software Engineering

Unfortunately the work is poorly presented and needs a deep revision of the data and much better presentation in the paper:

Extensive English improvement is needed

The abstract is too large and too many introductory phrases. Authors should be more straightforward (the audience has already read the title and know the importance of the topic). Include in the abstract the main conclusions of the survey

Prediction model is one of the most important techniques,
=>
Prediction model is an important technique,

"We found that prediction models have been applied to various software engineering tasks and we grouped them to 11 main research topics," Why those 11?

"There have been a large number of related studies in the last decade." => (PROVIDE REFERENCES, PLEASE)

There are broken references: "Section ??"

Introduction section is too short

"Prediction models have been applied to various tasks in software engineering." => REFERENCES NEEDED

The formulae in 2.3.X and the table do not provide interesting information to the reader. Authors should summarize the important and and basis of the methods commented

Authors do not justify why they chose those 3 conferences and 2 journals, there are many other interesting ones. There is also grey literature, snowball search, ...

Paragraphs in section 4.X are too large to be comfortable to read. Authors should divide them into several paragraphs and make an effort for them to be properly interconnected.

The same applies for 5.X, 6.X, 7.X, 8.X, 9.X, 10.X, 11.X, 12.X, 13.X and 14. Additionally a brief paragraph explaining the context of the feature and introducing the section would be appreciated by the reader

Authors claim that "In addition, datasets of many projects are not totally available due to the privacy 1062 policy," I totally agree. But to keep coherence they should share their data in a public website:  figshare, zenodo, etc

Conclusions are rather short. Authors should make an effort to make a briefing of the whole work in the paper: method used to research, findings, and future work.

Authors have to make an important effort to obtain interesting information from the data: first classify each paper according to different criteria (type of solution, proposal if it is implemented, if it was testing on actual data, etc etc), and show it in comprehensive tables and bubble/Venn diagrams. (see for example "Skill assessment in learning experiences based on serious games: A Systematic Mapping Study")

Reviewer 2 Report

This paper aims to create a comprehensive taxonomy of prediction models in software engineering. The paper postulates that this can be achieved through a comprehensive literature review inspired by commonly accepted principles of systematic literature reviews. Thus, the main intended contribution of this paper is the taxonomy itself, and as such the paper does not pose any research questions. The taxonomy contained in the paper also leads to the identification of challenges and directions for future research.

The paper needs some restructuring to fully encompass its true value. As it stands, the title can be misinterpreted as it suggests that the main contribution of this paper is the systematic literature review, which instead constitutes less than 10% of the paper. If the authors wish to keep the title, they will need to increase the information provided. In particular, they will need to add sections on research questions, paper quality evaluation, threats to validity, and results validation, which are all required by the guidelines the authors refer to (e.g., B. Kitchenham). Additionally, it is important to describe how the guidelines were applied in this case, and if any exceptions were made.

Section 2 needs to be justified, as it seem to stem from the systematic literature review, but is appears before such review is described and its results presented. It would be useful to understand if this collection of algorithms is presented "a priori," that is, as knowledge the authors acquired before they conducted the systematic literature review, and if so, where this has been derived from. and whether or not the algorithms described are in fact used in the papers listed in the, literature review. Are there also other algorithms found in the papers consulted, and if so what are they and why are they not included in this section?

The taxonomy is also slightly problematic, because it is not very clear how such a taxonomy was derived. Similarly, it would be important to clarify whether or not the knowledge on prediction models in Section 2.1 applies to all the papers in the literature review. These could all be part of the research questions for the systematic literature review.

The paper also mentions "other prediction models," but it does not discuss them further. Perhaps this is an oblique reference to the research directions described in Section 15.2. If so, it is important to establish that link explicitly as it is a contribution of the paper.

Challenges and directions also are said to be emerging from the papers in the review. However, there is no explicit link or citation. It would be important to either specifically cite the paper(s) that reference such challenges and directions; or, if these are the author's original ideas, they should be qualified as such. This could be accomplished by renaming Section 15 to "Discussion" or similar, and by clearly stating that these findings are the authors' and are based on the papers in the review.

Finally, the paper would be improved by a stylistic review to improve its readability and create flow between the sections. Comprehensive editing will accomplish this.

These suggestions add up to a major revision. The value this paper offers, however, would warrant such a revision if the authors were willing to consider it.

Round 2

Reviewer 1 Report

Authors have made a significant contribution to the paper but the main issue has not been address.

The paper just analysis the contributions in the different fields. But authors have to make a big effort in making the data synthesied for the readers.

There is no research question (RQ), and the paper is misleading. There is no "big picture" of the state of the art, just challenges and directions.

The analysis of the results (that are not available as open-data set) is limited. Saying that there is a paper is not enough. Authors should make a systematic approach and check different aspects of the proposal and make interesting charts and hypothesis contrast that can answer to RQ and help the reader draw results

There is "threats to validity" section

I encourage authors to include their resulting dataset as open-data in a public repository (figshare, zenodo, etc)

Authors can check the propals in many other SLR-based papers, like:

* Serious games in science education: a systematic literature
review

* Systematic Literature Review (SLR) on the application of serious
games in basic science courses for the virtual modality as a strategy
to improve the student retention rate

* Web-Based Serious Games and Accessibility: A Systematic Literature Review

* A systematic literature review on serious games evaluation: An application to software project management

Reviewer 2 Report

The revisions highlight the contribution this paper provides and are sufficient to warrant publication. A final editing pass is recommended to improve spelling (e.g., “Fundation”) and to adjust some phrases (e.g. “conduct a taxonomy”).

Author Response

We have done a final editing and adjust some phrases. Thank you very much for your advice. 

Round 3

Reviewer 1 Report

Authors have made a significant contribution to the paper but the main issue has not been address.

The paper just analysis the contributions in the different fields. But authors have to make a big effort in making the data synthesised for the readers.

Just adding research question (RQ) at the end of the paper does not align the paper with them.

The analysis of the results (that are not available as open-data set) remains very limited.

I encourage authors to include their resulting dataset as open-data in a public repository (figshare, zenodo, etc)

I did not recommend authors to add references to other SLR-based papers. I added some paper to help authors idintify their proposal's limitation

Reviewer 2 Report

The paper is ready for publication, with perhaps some very minor editing/spell check.